# RNA Sequencing Reveals the Potential Adaptation Mechanism to Different Hosts of *Grapholita molesta*

**DOI:** 10.3390/insects13100893

**Published:** 2022-09-30

**Authors:** Dongbiao Lü, Zizheng Yan, Di Hu, Aiping Zhao, Shujun Wei, Ping Wang, Xiangqun Yuan, Yiping Li

**Affiliations:** 1Key Laboratory of Integrated Pest Management on Crops in Northwestern Loess Plateau, Ministry of Agriculture, College of Plant Protection, Northwest A&F University, Yangling, Xianyang 712100, China; 2Institute of Plant Protection, Beijing Academy of Agriculture and Forestry Sciences, 9 Shuguanghuayuan Middle Road, Haidian District, Beijing 100097, China; 3Department of Entomology, Cornell University, Geneva, NY 14456, USA

**Keywords:** *Grapholita molesta*, host species, RNA sequencing, adaptation, trypsin

## Abstract

**Simple Summary:**

The oriental fruit moth *Grapholita molesta* is an important worldwide fruit tree pest. Its hosts mainly include peach, pear, apple, plum and other Rosaceae fruit trees. In order to explore the mechanism of host adaptation, it is necessary to comprehensively study the effects of feeding and the differences in transcriptional levels of *G. molesta*. In this study, the growth and development parameters, protease activities, midgut transcriptome analysis and key gene validation of *G. molesta* fed on five hosts and artificial diet were determined. This study indicated that the significant differences in the growth and development parameters and protease activities, transcriptome of KEGG enrichment and WGCNA analysis of *G. molesta* fed on different hosts. The transcriptional level of the trypsin gene was consistent with that of qRT-PCR. The trypsin gene played an important role in feeding different hosts of *G. molesta.* We revealed the adaptive mechanism to different hosts of *G. molesta* in various aspects and laid a solid foundation for further exploring the molecular mechanism of host adaptation.

**Abstract:**

*Grapholita molesta* is an important fruit tree worldwide pest which feeds on hosts extensively and does serious harm. In this paper, the growth and development parameters and protease activities of *G. molesta* fed on different hosts were compared. Using Illumina RNA sequencing technology, 18 midgut samples from five different hosts (apple, pear, plum, peach and peach shoots) and artificial diet were sequenced and compared with the reference genome, resulting in 15269 genes and 2785 predicted new genes. From 15 comparative combinations, DEGs were found from 286 to 4187 in each group, with up-regulated genes from 107 to 2395 and down-regulated genes from 83 to 2665. KEGG pathway analysis showed that DEGs were associated with amino acid metabolism, starch and sucrose metabolism, carbohydrate metabolism, and hydrolase activity. A total of 31 co-expression gene modules of different hosts were identified by WGCNA. qRT-PCR showed that the expression pattern of the trypsin gene was consistent with RNA sequencing. In this study, growth and development parameters, protease activity, DEGs, enrichment analysis and qRT-PCR were combined to reveal the adaptation process to different hosts of *G. molesta* in many aspects. The results of this study provide a basis for further exploration of the molecular mechanism of host adaptation of *G. molesta.*

## 1. Introduction

The oriental fruit moth, *Grapholita molesta* Busck (Lepidoptera: Tortricidae), is an important fruit tree worldwide pest, causing serious damage to stone fruit and pome fruit in Asia, Europe, America, Africa and parts of Australia, including peach, pear, apple, plum and other Rosaceae fruit trees [1,2,3]. It has a wide host range; the larvae not only feed on the leaves and shoots, but also on the fruit, causing serious economic losses [4]. The main reason for the perennially rampant plague is the multi-host damage, a characteristic of *G. molesta* which has multiple generations every year, and the damage rate can reach 70–80% [5]. The hazard scope and severity that *G. molesta* causes have aroused great attention to the potential of this species to infest fruit trees worldwide, threatening the fruit quality and the potential impact on the economy and the environment [6]. Biology is conducive to deepening people’s understanding of insect pests, for in-depth exploration of the mechanism of *G. molesta*’s adaptability to different hosts, and for providing theoretical support for future control of *G. molesta*.

The transcriptome is the bridge between genome and proteome information and the biological function of genes. RNA sequencing has been widely used in transcriptome and genome analysis in recent years [7]. Transcriptomic techniques have been used to study the larval–pupal metamorphosis, genome-wide identification of serine proteases and midgut response to toxins of *Plutella xylostella* [8,9,10]. A combination of epidermal transcriptome and expression dynamics of target genes was used to study the larval–pupal metamorphosis of *Spodoptera litura* and *Bombyx mori* [11,12]. Transcriptomic analysis was used to study the physiological response to different host plants of *Helicoverpa armigera* and against baculovirus infection by oxidative stress of *Cnaphalocrocis medinalis*, as well as the screening of potential control target genes of *Spodoptera exigua* and comparative analysis of microRNAs at different developmental stages of *Manduca sexta* [13,14,15,16]. RNA sequencing has also been used in studies of *Bradysia odoriphaga*, *Panonychus ulmi*, *Cimex lectularius*, *Tetranychus cinnabarinus* and aphids [7,17,18,19,20]. Transcriptomic studies of *G. molesta* include sequencing analysis of sex pheromone glands, antennae, different developmental stages and microRNAs [21,22,23,24]. However, there is no study on transcriptome analysis of the different hosts fed on by *G. molesta*.

Host plants are important factors affecting insect growth, reproduction and population dynamics. The use of multiple hosts is the main cause of the serious harm of *G. molesta*. Najar-Rodriguez et al. found that peach was the main host, while apple and pear were the secondary hosts [25]. The physiological function of the larvae feeding on apple or pear was significantly different from that of peach, and apple was more conducive to population expansion than pear, which was consistent with the results of Myers et al. [25,26]. Du et al. used peach and pear collected at different times in the growing season to raise *G. molesta* and observed its development and fecundity [1]. The results showed that the development rate on peach fruit and shoots was faster than that on pear, and the pupal weight on peach fruit was significantly higher than that on shoots [1]. Yuan et al. found that there were significant differences in the intestinal microflora structure among *G. molesta* fed on artificial diet, apple, pear, pea, peaches, plum and peach shoots [27]. *G. molesta* feed on different hosts that cause great changes in physiological functions. These results provide a new idea for the study of host adaptation of *G. molesta*. Transcriptome analysis of *G. molesta* fed on different hosts will provide reliable theoretical support for the study of host adaptation of *G. molesta* [6].

In this study, we compared the growth and development parameters and the activities of three proteases of *G. molesta* fed with different hosts. Illumina RNA sequencing was used to sequence 18 samples fed on five hosts and artificial diet, and the clean reads were mapped to the reference sequence to detect differential expression of *G. molesta* feeding on different hosts and artificial diet. The differentially expressed genes (DEGs) between feeding hosts were analyzed, and Gene Ontology (GO) annotation showed relevant biological and molecular functions. Using pairwise comparisons and weighted gene co-expression network analysis (WGCNA), we identified modules of co-expressed genes and candidate hub genes for different hosts. In addition, quantitative real-time PCR (qRT-PCR) showed that the expression pattern of the screened trypsin gene was consistent with RNA sequencing. This work will help to elucidate in detail the complexity of the transcriptome of *G. molesta* and provide valuable transcriptomic resources for further study of host adaptation mechanisms.

## 2. Materials and Methods

### 2.1. Insect Culture and Host Category

Laboratory trials were carried out at the College of Plant Protection, Northwest A&F University, Yangling, Shaanxi, China. *G. molesta* larvae were obtained from colony maintained on an artificial diet and were kept at 25 ± 1 °C with 70 ± 5% relative humidity under a light/dark photoperiod of 15:9 h [2]. In the new generation, eggs from the same batch were collected. When the eggs hatched, the larvae were transferred to different hosts and artificial diet (AD) for feeding. These hosts included apples (*Malus pumila* M, cultivar Fushi, AP), pears (*Pyrus bretschneideri* R, cultivar Dangshan, PR), plums (*Prunus salicina* L, cultivar Jujin, PL), peaches (*Prunus persica* L, cultivar Qingyan, PC) and peach shoots (*Prunus persica* L, cultivar qingyan, PS). The components of the artificial diet were formulated as described by Su et al. [28]. Peach shoots were picked from the tender shoots of peach trees, while the other 4 hosts were all fruits purchased locally (Yangling, Shaanxi, China).

### 2.2. Growth and Development Parameters of Feeding on Different Hosts

Newly hatched eggs were transferred to different hosts as described above, and larval development and survival were recorded daily until pupation and emergence of pupae. On the second day after pupation, pupae were weighed and classified as male and female. After pupae emerged, adult pairs (♀:♂ = 1:1) were placed in disposable paper cups (fed with 10% honey water), and the paper cups were changed daily to record the number of eggs laid by single female and the developmental period of adult. In total, 100 larvae of each host were used in triplicate. The R_0_ (net proliferation rate), T (average generation period), r_m_ (intrinsic rate of natural increase), λ (finite rate of increase) and t_d_ (population doubling time) were calculated according to the following formula. To compile the population life table of *G. molesta* [29,30,31]:R_0_ = ∑L_x_ m_x_(1)
T = ∑xL_x_ m_x_/R_0_(2)
r_m_ = lnR_0_/T(3)
Λ = e^rm^(4)
T = ln2/r_m_(5)
x represents the time interval (day), L_x_ means the survival probability of female adults on day x after hatching, and m_x_ refers to the average fecundity of female adults on day x after hatching.

### 2.3. Midgut Protease Activity of Larvae Fed on Different Hosts

The larvae were dissected on ice and washed with 0.15 mol/L NaCl solution. The midgut and contents of the 2nd, 3rd and 4th instar larvae were harvested and placed in NaCl solution and homogenized in an ice bath. The homogenate was centrifuged at 12,000 rpm for 5 min at 4 °C, and the supernatant was used as the midgut enzyme extract for detection. The total soluble protein content in the enzyme solution was determined by 2-D Quant Kit [32,33]. The activity of each protease was determined according to the method of Zhao et al. [34]. The activities of total protease, high-alkaline trypsin and low-alkaline trypsin were measured at wavelengths of 415 nm, 405 nm and 247 nm using AZOCASEIN (Sigma Chemical Co., St. Louis, MO, USA), BAPNA (Nα-Benzoyl-DL-argininep-nitroanilide) (Sigma Chemical Co., St. Louis, MO, USA) and TAME (Nα-P-Tosyl-L-arginine methyl ester hydrochloride) (Sigma Chemical Co., St. Louis, MO, USA), respectively.

### 2.4. Sample Collection and RNA Extraction

The 4th instar *G. molesta* larvae fed on different hosts were collected as experimental samples. The larvae were sterilized and cleaned with 0.5% sodium hypochlorite, 70% ethanol and sterile water, and the midgut was removed and stored in pre-cooled phosphate buffered saline (PBS, pH 7.4). The experiment involved samples fed on 5 different hosts and AD. A sample of 30 midguts was used, with 3 replicates per sample. Total RNA was extracted from the above samples using Trizol Reagent Kit (Invitrogen, Carlsbad, CA, USA). RNA degradation and contamination were detected by electrophoresis using ribonuclease (RNase)-free agarose gel. The purity and concentration of RNA were measured by a NanoDrop 2000 spectrophotometer (Agilent Technologies, Santa Clara, CA, USA) to obtain OD 260/280 and OD 260/280 values. RNA integrity and total amounts were accurately measured using Agilent 2100 Bioanalyzer (Agilent Technologies, Santa Clara, CA, USA). The total RNA was then stored at 80 °C for subsequent experiments. For Illumina RNA Sequencing, 18 libraries were prepared from RNA samples from 5 different hosts and AD.

### 2.5. Library Preparation and Sequencing

The mRNA was purified from total RNA using poly-(T) oligo-attached magnetic beads. Then the enriched mRNA was fragmented into short fragments using divalent cations under elevated temperature in a strand synthesis reaction buffer, and cDNA was synthesized using a random hexamer primer and M-MuLV reverse transcriptase (RNase H). The cDNA fragments were purified with a Qiaquick PCR Extraction Kit (Qiagen, Venlo, The Netherlands). After adenylation of 3′ ends of DNA fragments, an adaptor with a hairpin loop structure was used for ligation to prepare for hybridization. In order to select cDNA fragments of preferentially 370–420 bp in length, the library fragments were purified with AMPure XP system (Beckman Coulter, Beverly, CA, USA). PCR was performed, the products were purified (AMPure XP system), and the library quality was assessed on the Agilent Bioanalyzer 2100 system. Then, Qubit 2.0 Fluorometer (Life Technologies, Waltham, MA, USA) was used for initial quantification, diluted to 1.5 ng/μL. The Agilent Bioanalyzer 2100 (Agilent Technologies, Santa Clara, CA, USA) was used to measure the insert size length of the library, and qRT-PCR (Vazyme) accurately quantified the effective concentration of the library, making the effective concentration higher than 2 nM to ensure the quality of the library. The clustering of the index-coded samples was performed on a cBot cluster generation system using TruSeq PE Cluster Kit v3-cBot-HS (Illumina, Albany, NY, USA). After cluster generation, the library preparations were sequenced on an Illumina NovaSeq platform generated by Novogene Co., Ltd. (Beijing, China).

### 2.6. Quality Control and Normalization

Raw data were first processed through in-house Perl scripts. The image data of the sequence fragments measured by the high-throughput sequencer were converted into sequence raw data by CASAVA base recognition. The quality control software was Fastp (version 0.19.7, parameters: -g -q 5 -u 50 -n 15 -l 150) (https://github.com/mskcc-cwl/fastp_0.19.7). The raw data obtained by sequencing included a small number of reads containing adapter or low quality. Raw data were filtered by removing reads containing adapter, uncertain bases and low quality. Clean data were calculated for Q20, Q30 and GC content. All subsequent analyses were performed with high quality clean data to obtain final transcripts for subsequent analyses.

### 2.7. Transcriptome Sequence Mapping to the Reference Genome

Clean reads after quality control sequencing were compared with the reference genome. HISAT2 (v2.0.5) (http://daehwankimlab.github.io/hisat2) was used to quickly and accurately compare clean reads with the reference genome, and a database of splices and links was generated based on gene model annotation files to obtain the location information of reads on the reference genome [35].

### 2.8. Quantification of Gene Expression Levels

Statistical and quantitative analysis of the number of reads covered by each gene were performed according to the location information of the gene comparison to the reference genome of *G. molesta*. Based on the transcriptome sequence obtained from the above quality control, the readings mapped to each gene were calculated using featureCounts (1.5.0-p3) (https://www.researchgate.net/publication/258524029_FeatureCounts_An_efficient_general_purpose_program_for_assigning_sequence_reads_to_genomic_features) and normalized to the FPKM value of the transcript, the expected number of fragments per million base pairs of sequences [36,37,38]. The software edgeR (3.5.0) (https://bioconductor.org/packages/release/bioc/html/edgeR) was used to map and analyze the clean reads obtained by sequencing [39].

### 2.9. Differentially Expressed Genes (DEG)

After gene expression quantification, normalization and statistical model hypothesis testing probability (*p*-value) were performed on the original read count. Multiple hypothesis testing was performed to correct the FDR value (padj is the common form) [39,40]. DESeq2 (1.20.0) was used to analyze the differential expression between groups [41]. Transcriptome analysis was conducted for many genes, which leads to the accumulation of false positives. The higher the number of genes, the higher the degree of false positive accumulation in hypothesis testing. In order to control the proportion of false positives, padj was used to correct *p*-value of hypothesis testing [42]. At the same time, to meet the threshold standard for screening differential genes of |log2(foldchange)| ≥ 1 and padj ≤ 0.05.

### 2.10. GO and KEGG Pathway Enrichment Analysis

We classified the genome annotation information and mapped the DEGs to the Gene Ontology (GO. Available online: http://www.geneontology.org) (accessed on 14 November 2021) for enrichment analysis. GO function enrichment was considered to be a differentially expressed gene with padj less than 0.05 as the threshold of significant enrichment. The result of enrichment analysis was that all DEGs, up-regulated DEGs and down-regulated DEGs of each differential comparison combination were enriched. Kyoto Encyclopedia of Genes and Genomes (KEGG. Available online: http://kobas.cbi.pku.edu.cn/home.do) (accessed on 14 November 2021) is a comprehensive database integrating genomic, chemical and systematic functional information. The threshold of KEGG pathway enrichment was padj less than 0.05. ClusterProfiler (3.8.1) software was used for GO functional enrichment analysis and KEGG pathway enrichment analysis of DEGs. In this study, there are 5 hosts and AD, 15 comparison combinations, including 3 types of feeding foods (artificial diets, fruits and peach trees shoots). AD and PS were used as representatives of artificial diets and peach trees shoots, respectively. PL and PR were selected as representatives of fruits for combinatorial analysis.

### 2.11. WGCNA

WGCNA (weighted gene co-expression network analysis, v1.47. Available online: https://horvath.genetics.ucla.edu/html/CoexpressionNetwork/Rpackages/WGCNA) (accessed on 25 November 2021)uses high-throughput gene messenger RNA (mRNA) expression data to construct co-expression network hierarchical cluster trees, measure whether genes have similar expression patterns, and describe gene association patterns among different samples. After the data were imported into the software, the gene network was assumed to follow the scale-free distribution, and the Pearson coefficient between genes was calculated by using the gene co-expression correlation matrix and the adjacency function formed by the gene network, and then the hierarchical clustering tree was constructed. The different branches of the cluster tree represent different genetic modules, which are associated with feeding on different hosts to obtain important modules related to the midgut digestion of *G. molesta*. KEGG pathway enrichment analysis in each module was conducted to analyze the biological functions of modules.

### 2.12. Quantitative Real-Time PCR (qRT-PCR)

The DEGs were obtained based on the FPKM value of each host and pairwise comparison results, from which trypsin genes were screened (Appendix A). Primer Premier 5.0 software was used to design the primers for fluorescent quantitative PCR (Appendix A). According to the above methods, the 4th instar larvae of *G. molesta* were collected, and RNA was extracted. The cDNAs were synthesized using a First Strand cDNA Synthesis Kit (TaKaRa) following the manufacturer’s protocol. Quantitative PCR analysis was performed by using a 2×ChamQ SYBR qPCR Master Mix (Vazyme). The PCR conditions were as follows: initial denaturation at 95 °C for 30 s, followed by 40 cycles of 95 °C for 10 s and 60 °C for 30 s. The relative mRNA levels of gene expression were calculated using the 2^-^^△△Ct^ method. Internal reference genes were *β-actin* and *EF-1α* (GenBank: JN857938 and KT363835). All data included 3 biological replicates.

### 2.13. Statistical

With statistical analysis based on the assumptions of normality and homogeneity of variance, ANOVA followed by Tukey’s HSD test was used for data analysis. All valid data were analyzed using SPSS Statistics 26.0 (Available online: https://www.ibm.com/products/spss-statistics) (accessed on 5 March 2021). Differences were considered significant when * *p* < 0.05.

## 3. Results

### 3.1. Effects of Different Hosts on Growth and Development

Feeding on hosts and AD can carry out the complete life cycle, but the parameters were different. The developmental duration on different hosts and AD, including larval stage, pupal stage, adult stage, larval adult stage and the next generation egg stage were significantly different (Table 1). The larval stage, pupal stage and egg stage on PS were the longest. The larval–adult stage on PL and PS were 37.56 ± 0.52 and 37.55 ± 0.59, respectively. The larval and egg stages on PC were the shortest, then the adult stage was the longest. Larval survival rate, pupation rate, pupal emergence rate and fecundity of feeding on different hosts varied significantly (Table 2). The larva survival rate, pupation rate and fecundity on PS were the lowest, then pupal weight was the largest. There was no significant difference in larval survival rate between PC and AP, with PC being the largest, and pupation rate being the same. AP had the highest Emergence rate and fecundity. There were significant differences in life table parameters (R_0_, T, r_m_, λ, t_d_) feeding on different hosts (Table 3). The R_0_ on AP was the largest, and the minimum on PS was 95.13 ± 0.03. The T on AD and AP were 38.52 ± 0.29 and 38.83 ± 0.09, the r_m_ were 0.13 ± 0.02 and 0.13 ± 0.01, the λ were 1.22 ± 0.01 and 1.23 ± 0.01, and the t_d_ were 5.95 ± 0.03 and 5.90 ± 0.03, respectively. The t_d_ on PS was the largest.

### 3.2. Effects of Different Hosts on Protease Activity in Larval Midgut

The activities of the three proteases reached the maximum at 4th instar (Figure 1). The total protease activity of feeding on different hosts was between 0.2510 ± 0.0093 and 0.4207 ± 0.0051 OD/min/mg (Figure 1A). The total protease activity on PS was the lowest among all hosts. At each instar, no significant differences were observed in other hosts except PS. The high-alkaline trypsin activity was between 0.3965 ± 0.0056 and 0.8426 ± 0.0088 OD/min/mg (Figure 1B). The high-alkaline trypsin activity on PS was the lowest among all hosts, which was significantly different from that of other hosts. AD, PC and AP are larger, followed by PR and PL. The low-alkaline trypsin activity was between 0.5195 ± 0.0026 and 0.6511 ± 0.0048 OD/min/mg (Figure 1C). The activity on PS at 4th instar was significantly lower than that of other hosts.

### 3.3. Summary Transcriptome Sequencing Data

The 18 midgut samples of *G. molesta* larvae were collected for high-throughput RNA sequencing. Through quality control, the raw reads were distributed between 38,070,174 and 61,963,606. The clean reads obtained after filtering were distributed between 36,448,254 and 60,618,820. The percentage of valid reads ranged from 90.73% to 98.20%. Q20 ranged from 95.02% to 98.86%. Q30 was 88.35% to 96.34%. GC base content ranged from 46.59% to 57.98% (Appendix A). The sequencing volume and depth of samples were large enough. The sampling and sequencing operation of each treatment group was appropriate, the sequencing integrity was good, and reliable data valuable for further analysis were obtained.

### 3.4. Map and Annotation Analysis

By aligning the above 18 transcriptomes with the reference genome, we obtained details of the read map. The proportion of total reads that could be mapped to the genome were between 77.14% and 89.33%; the reads that could be uniquely mapped were 69.03% to 85.68%. Read1, read2, positive and negative that were mapped to the genome were almost all above 35.00%. Pairs of reads mapped to the genome were between 61.35% and 79.39% (Appendix A). A total of 18,054 transcript sequences were annotated after transcript and genome mapping. A total of 15,269 genes were mapped, and 2785 novel genes were predicted.

### 3.5. Analysis of Gene Expression Levels

FPKM was used to correct the sequencing depth and gene length. The gene expression levels of different hosts were compared by analyzing the boxplot of gene expression level distribution, the FPKM density distribution and the violin plot of FPKM distribution. The FPKM of insects feeding on different hosts can be analyzed statistically and qualitatively from the genetic height and overall level. The results showed that the density and number of genes were in general basically similar (Figure 2, Appendix A). The correlation coefficients (R^2^) of samples within and between groups were calculated according to the FPKM values of all genes in each sample. It was found that the R2 of AD, PC, PR and PL were all greater than 0.83, and even that for PL was greater than 0.93, indicating good biological repetition among different hosts. However, the R^2^ between PS and AP were slightly lower (0.671–0.815), which may be that the adaptability of insects to hosts is different due to the difference in hosts, and also by incomplete cleaning of impurities in sample processing or mechanical errors in instrument operation (Appendix A). Principal component analysis (PCA) was used to assess inter-group differences and intra-group repeatability by dimensionality reduction and principal component extraction of a large number of genetic variables. A linear algebra calculation method was adopted to compare the similarity of the main components of feeding different hosts, and the differences in multiple groups of data were reflected by a 2-dimensional coordinate graph (Figure 3). In the scatterplot, the X-coordinate PC1 explained 27.87% of the variation in the data, while the Y-coordinate PC2 explained 16.27% of the variation. As shown in the figure, each host sample occasionally intersects slightly but is basically clustered separately.

### 3.6. Differentially Expressed Gene (DEG) Analysis

After quantification of gene expression and differential screening of the transcriptome data of the larvae midgut of *G. molesta* fed on 5 different hosts and AD, predicted lncRNA transcripts were screened from differentially expressed genes for differential lncRNA analysis, and the differences between up-regulated and down-regulated lncRNAs in pairs of different comparison combinations were determined. We found 286 to 4187 DEGs in each group from 15 comparison combinations of 5 hosts and AD, the up-regulated genes ranging from 107 to 2395 and the down-regulated genes ranging from 83 to 2665. The DEGs of AP vs. PL and PC vs. PS were 4187 and 286, the up-regulated genes of AD vs. AP and AP vs. PS were 2395 and 107, and the down-regulated genes of AP vs. PL and PR vs. PC were 2665 and 83, respectively (Figure 4). Five hosts were compared with AD as the control. In the comparative combinations of AD vs. AP, AD vs. PR, AD vs. PC, AD vs. PL and AD vs. PS, 3650, 1842, 2784, 3944 and 815 assembled transcripts were obtained, respectively. Of these, 76 were common transcripts. Uniquely expressed transcripts were 1782, 238, 583, 1577 and 127, respectively (Figure 5).

### 3.7. GO and KEGG Pathway Enrichment Analyses

Data from the transcriptome were used to analyze the DEGs of feeding on different hosts (AP, PR, PR, PC and PS) and AD of *G. molesta*. The DEGs were obtained by pairwise comparison of these 5 hosts and AD. A total of 7939 genes were annotated by GO functional classification. They belong to 3 categories and 46 branches, namely molecular functions, cellular components and biological processes. Molecular functions mainly include 10 functions such as binding, catalytic activity and transport activity. Cell components mainly include 16 functions such as cells, membranes and organelles. Biological processes mainly include metabolic processes, single-organism processes and cellular processes and 20 other functions (Figure 6).

There were 797 genes showing different expression levels between AD and PL, including 537 up-regulated genes. Functional enrichment results indicated that these up-regulated DEGs in GO terms were mainly related to the oxidation–reduction process, iron ion binding, heme binding, tetrapyrrole binding, oxidoreductase activity, cofactor binding, serine-type endopeptidase activity, serine-type peptidase activity, serine hydrolase activity, hydrolase activity and endopeptidase activity. They mainly belong to molecular functions and biological processes, respectively (Appendix A). These up-regulated DEGs were significantly involved in KEGG pathways including those related to tyrosine metabolism, glycine, serine and threonine metabolism, metabolism of xenobiotics by cytochrome P450, drug metabolism, biosynthesis of cofactors, ubiquinone and other terpenoid–quinone biosynthesis, starch and sucrose metabolism, biosynthesis of unsaturated fatty acids, ascorbate and aldarate metabolism, thiamine metabolism, pentose and glucuronate interconversions, peroxisome, propanoate metabolism, alanine, aspartate and glutamate metabolism, steroid biosynthesis, phenylalanine metabolism and beta-alanine metabolism (Figure 7A).

There were 1777 genes showing different expression levels between PR and PL, including 1246 up-regulated genes. Functional enrichment results indicated that these up-regulated DEGs in GO terms were mainly related to the oxidation–reduction process, proteolysis, extracellular regions, oxidoreductase activity, structural constituents of cuticle, iron ion binding, cofactor binding, heme binding, tetrapyrrole binding, serine-type endopeptidase activity, peptidase activity, serine-type peptidase activity, and serine hydrolase activity (Appendix A). These up-regulated DEGs were significantly involved in KEGG pathways including those related to tyrosine metabolism, glycine, serine and threonine metabolism, carbon metabolism, biosynthesis of unsaturated fatty acids, arginine biosynthesis, drug metabolism, biosynthesis of amino acids, lysosomes, pyruvate metabolism, the citrate cycle (TCA cycle), biosynthesis of cofactors, longevity regulating pathways, ascorbate and aldarate metabolism, Toll and Imd signaling pathways, metabolism of xenobiotics by cytochrome P450, thiamine metabolism, nitrogen metabolism, arginine and proline metabolism, and alanine, aspartate and glutamate metabolism (Figure 7B).

There were 739 genes showing different expression levels between PL and PS, including 621 down-regulated genes. Functional enrichment results indicated that these down-regulated DEGs in GO terms were mainly related to the oxidation–reduction process, the alpha-amino acid metabolic process, extracellular regions, structural constituents of cuticle, iron ion binding, oxidoreductase activity, cofactor binding, heme binding, tetrapyrrole binding, structural molecule activity, hydrolase activity and coenzyme binding (Appendix A). These down-regulated DEGs were significantly involved in KEGG pathways, including those related to pentose and glucuronate interconversions, metabolism of xenobiotics by cytochrome P450, drug metabolism, ubiquinone and other terpenoid–quinone biosynthesis, glycine, serine and threonine metabolism, tyrosine metabolism, longevity regulating pathways, cysteine and methionine metabolism, ascorbate and aldarate metabolism, lysosomes, retinol metabolism and biosynthesis of amino acids (Figure 7C).

There were 379 genes showing different expression levels between PR and PS, including 184 down-regulated genes. Functional enrichment results indicated that these down-regulated DEGs in GO terms were mainly related to iron ion binding, oxidoreductase activity, hydrolase activity, cofactor binding, heme binding, tetrapyrrole binding, transmembrane transporter activity, transporter activity and NAD binding (Appendix A). These down-regulated DEGs were significantly involved in KEGG pathways including those related to ubiquinone and other terpenoid–quinone biosynthesis, drug metabolism, metabolism of xenobiotics by cytochrome P450, ascorbate and aldarate metabolism, pentose and glucuronate interconversions, cysteine and methionine metabolism, carbon metabolism and glutathione metabolism (Figure 7D).

### 3.8. Gene Co-Expression Network Interactions

WGCNA was used to analyze the association between co-expressed gene modules and to reveal the key genes related to feeding and digestion of different hosts of *G. molesta*. By selecting an appropriate soft threshold, the correlation between gene expression levels was obtained, and a gene clustering tree was constructed (Appendix A). Dynamic cutting and merging of trees were used to obtain the total co-expression modules of different genes (Figure 8A). By clustering the distance of module characteristic genes (ME), the module network tree was constructed. A total of 31 different gene modules was identified (Figure 8B). In order to quantify the co-expression similarity of the whole module, the adjacency correlation of 31 characteristic genes based on a heat map was calculated. Gradually saturated green and red indicate high co-expression correlation, and ME represents unused modules (Figure 8C). MEgreenyellow and MElightcyan were expressed in PL, MEpaleturquoise in PS, MEblue in AP, MEcyan, MEskyblue, MEgreenyellow, and MEyellow in PC, MEdarkred and MEred in PR (Figure 8D).

### 3.9. Quantitative Real-Time PCR (qRT-PCR) Analyses

Quantitative real-time PCR was performed to validate 9 genes obtained from RNA-sequencing data (Gene IDs were gm_26520-RA, gm_38295-RA, gm_04701-RA, novel.798, gm_11542-RA, gm_03663-RA, gm_06101-RA, gm_35262-RB and gm_35484-RH). As expected, qRT-PCR and RNA-sequencing analyses of these 9 genes showed similar differential expression patterns, proving the reliability of our data and analyses (Figure 9).

## 4. Discussion

The growth, development, reproduction and population dynamics of insect populations are affected by host plants [43]. In this study, we investigated the growth, development and reproduction of *G. molesta* feeding on AD, PC, AP, PS, PR and PL, and found significant differences in the effects of host. The larval stage on PC was the shortest, and the pupal stage on AP was the shortest. The adult stage on PC was the longest, followed by AP. On the contrary, PS has the longest larval and pupal stages, while the shortest adult stage. The larval survival rate and pupation rate of PC and AP were the highest. The larval stage of PC is shorter than that of AP, which is consistent with the conclusion of Myers et al. and Najar-Rodriguez et al. [25,26]. Najar-Rodriguez et al. showed that pupal weight was positively correlated with fecundity, while the results of this study did not show a correlation, which may be due to the decrease in fecundity in the population reared in the laboratory due to the relatively mild and stable environment [25]. As important parameters in the life table, R_0_ and r_m_ can reflect the reproductive ability and host adaptability of insect populations in specific environments [44]. AP has the largest R_0_, the shortest T, and the shortest t_d_, followed by PC. Considering the emergence rate, reproduction rate, R_0_ and T, AP and PC were the most suitable hosts, and AD was also a kind of food for the normal growth and development and population maintenance of *G. molesta*. The results of this study are similar to those of Myers et al. [26,45]. The activities of total protease, high-alkaline trypsin and low-alkaline trypsin of PS were significantly lower than those of other hosts when fed on different hosts, which may be because PS belonged to shoots and had high cellulose content and low protein content. The activities of the three proteases were the highest in the 4th instar, and the differences between hosts were significant. The results of this study are similar to those of Zhao et al. on *Plutella xylostella* and Lv et al. on *Illiberis pruni* [32,33].

High-throughput transcriptome sequencing technique is a common method to study functional genes. The technology is mature and reliable and has been widely used to explore insect genes, such as *S. litura*, *B. mori*, *P. xylostella* and *G. molesta* [8,9,11,19,21,22,23,24]. Jung et al. on sex pheromone glands, Li et al. on antennae and Guo et al. on eggs, larvae, pupae and adults fed with artificial diet performed transcriptome sequencing analyses of *G. molesta* [21,22,23,24]. In combination with the growth and development parameters and protease activity characteristics, we describe a deep RNA-sequencing data set of the midgut of *G. molesta* larvae fed on five different hosts and AD. Overall, we found significant differences between samples fed on different hosts. When we set up different hosts, we deliberately set up two types of hosts and AD, namely fruit, and peach tree shoots. Interestingly, when we compared the artificial diet to fruit, fruit to fruit, and fruit to shoots, we found a result that matched our expectations. The expression trend of screened trypsin genes in all hosts was consistent with that of the transcriptome. This result makes the study even more significant.

According to the Pearson correlation heat map of each sample, the correlation coefficients of AD, PC, PR and PL were slightly higher than PS and AP. PC, PR and PL all belong to the fruit of Rosaceae and have similar preferences and adaptability with *G. molesta* [1]. AD is a mixture we made according to various nutritional needs, and its nutritional composition and feeding preference degree are similar to the above fruits, so the correlation between the four is very high. PS is a peach tree shoot, which contains a lot of cellulose, and its nutritional composition and fruit are very different. When PS is eaten by *G. molesta*, the individual insect has the adaptation process, due to the individual adaptability difference, so then the correlation also has that certain difference. Similarly, when eating AP, the correlation was slightly lower with some difference. The Myers et al. study showed that there were significant differences in growth and development of *G. mole**sta* fed on AP and PC [46]. When feeding on AP, the differences between individuals will be further amplified. When the attributes of feeding hosts differ, the correlation changes accordingly. According to PCA, it was found that the four hosts of PL, AP, PR and AD had good repeatability, and each host was clustered separately. The distributions of PC and PS are slightly dispersed, but the whole population can basically gather independently. The correlation between them is high, which may be caused by mechanical error. Interestingly, PC and PS are both parts of the peach tree. This problem is worth further exploration.

In the current study, RNA-sequencing analysis was conducted to clarify the adaptation process of different hosts at the molecular level of *G. molesta.* The GO and KEGG analyses can help us better understand the biological functions of DEGs, and thus the specific annotations and pathways that were significantly enriched in this study might play a crucial role in the process of the adaptation to feeding on different hosts by *G. molesta*. As expected, we found large numbers of DEGs were annotated and enriched in metabolization-related terms, suggesting significant transcript changes in digestion and metabolism during adaptation to different hosts of *G. molesta*. According to the number of DEGs compared pairwise, PC vs. PS had the least. Although one is the fruit and the other is the leaf, it may be because they both are parts of the peach tree that there is little difference. The AP vs. PS number were also small, consistent with our conjecture above, and these two hosts were not highly correlated with the other three hosts and AD, probably because they had many similarities.

By comparing transcriptome sequencing with reference genes, differentially expressed genes feeding on different hosts were obtained. When AD is prepared, preservatives such as ascorbic acid and methyl paraben (which are inherently toxic) are added to prevent decay and increase preservation time. Therefore, the expression levels of cytochrome P450 metabolism, drug metabolism, ascorbate and aldarate metabolism are found to be elevated for AD vs. PL [28]. In order to ensure the nutritional composition of AD and the demand by insects for sucrose, sucrose and starch must be added to AD. KEGG enrichment indicated that digestive enzymes enhanced its metabolism. The Su et al. study showed that sucrose is an essential nutrient of *G. molesta*, which provides energy for flight [28]. There were significant differences in larval and pupal periods, adult longevity and pupal weight. Kaufmann et al. reached the same conclusion in the study of *Scathophaga stercoraria* [47]. According to the effects of each component of AD and the conclusions of this study, the data analyzed by KEGG are valid and consistent with the research status. Research by Du et al. shows that pears are rich in amino acids and sugars, and are one of the hosts on which *G. molesta* is the most voracious [1]. Cytochrome P450 metabolism, drug metabolism and some amino acid metabolism have been described above and are among the processes necessary to maintain normal insect growth and development [28]. In PR vs. PL, glycolysis and gluconeogenesis expression were increased, which was consistent with the research status. In PL vs. PS and PR vs. PS, PS is the shoot with more cellulose and more secondary metabolites in shoots. Most of the secondary metabolites are toxic, so the biosynthetic expressions of cytochrome P450 metabolism, glutathione metabolism, drug metabolism, ubiquinone and other terpenoid–quinone biosynthesis are decreased, consistent with current research.

We conducted WGCNA feeding on five different hosts and AD, obtained 31 co-expression network modules, and found that each host had its own unique expression module. Only some modules were expressed in multiple hosts. The results of WGCNA were basically the same as those of KEGG, which are consistent with the research status. Combined with the above DEG results, the results of qRT-PCR and RNA sequencing of the screened nine trypsin genes were the same. The Kumar et al. study showed that qualitative changes in trypsin activity in the midgut of *Pieris brassicae* were related to physiological adaptation to transfer to different hosts [46,48]. Huang et al. studied the growth, size, development and survival rate of *Oedaleus asiaticus* feeding on *Cleistogenes squarrosa*, *Leymus chinensis*, *Stipa krylovii*, *Artemisia frigida* and *Caragana microphylla* [49]. Transcriptome analysis confirmed altered gene expression levels feeding on different hosts. Rivera-Vega et al. conducted RNA-sequencing analysis on the salivary glands of *Trichoplusia ni* feeding on cabbage, tomato, and pinto bean artificial diet and concluded that salivary glands play an important role in host selection and adaptability [50]. By analyzing the RNA-sequencing transcriptome of *Nezara viridula* fed on corn and mung bean, Canton et al. identified specific transcripts that are transcribed differently according to diet and across multiple tissues [51]. The conclusion of this study is similar to that of the above research: feeding on different hosts will cause adaptability differences in insects. This is mainly reflected in the differences in gene expression levels, and trypsin in this study is one of them.

## 5. Conclusions

In this study, the significant differences in the growth and development parameters and protease activities of *G. molesta* fed on different hosts, combined with the results of RNA-sequencing transcriptome analysis, revealed the adaptation mechanism to different hosts of *G. molesta*. A series of modules and genes related to host adaptation were screened by comparing DEGs and their corresponding molecular physiological pathways, KEGG enrichment analysis and WGCNA. Combined with qRT-PCR, the expression pattern of the trypsin gene was consistent with RNA sequencing. From the differences in growth and development parameters, protease activity characteristics, DEGs screening, enrichment analysis and qRT-PCR, the adaptation process of *G. molesta* to different hosts was comprehensively revealed. The results of this study lay a solid foundation for further exploration of the molecular mechanism of host adaptation of *G. molesta*.

## Figures and Tables

**Figure 1 insects-13-00893-f001:**
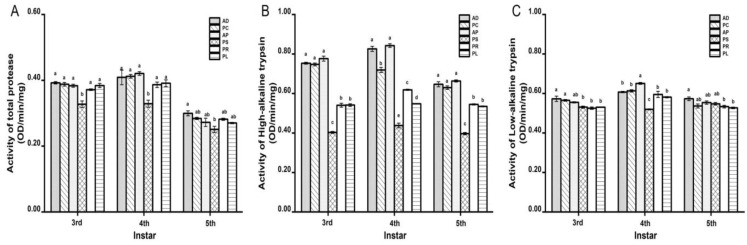
Effects of different hosts on the protease activity in the midgut of *G. molesta* larvae. (**A**) Total protease. (**B**) High-alkaline trypsin. (**C**) Low-alkaline trypsin. Different letters indicate significant differences among different hosts (*p* < 0.05).

**Figure 2 insects-13-00893-f002:**
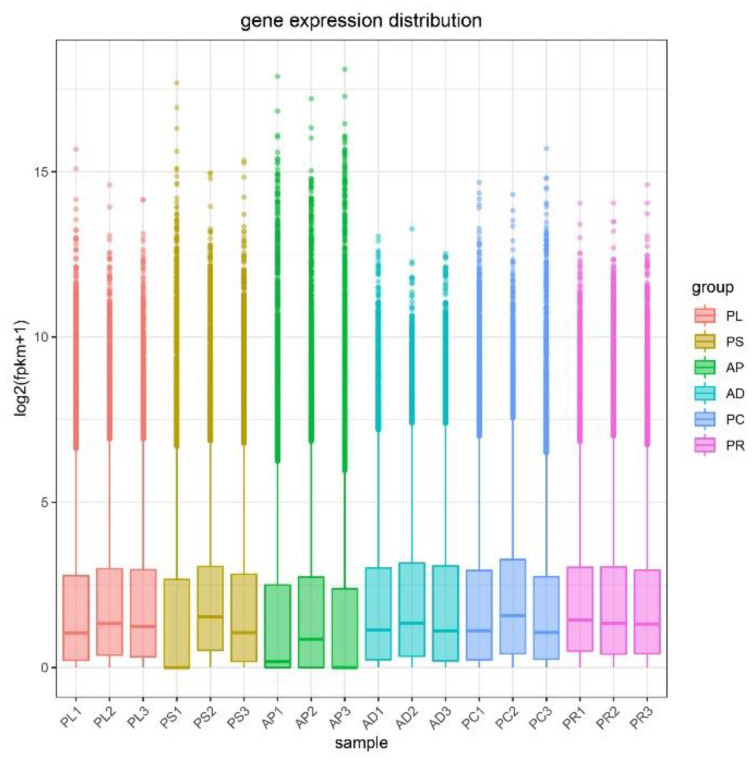
Gene expression distribution boxplot of each sample. The X-coordinate is the sample name, and the Y-coordinate is log2 (FPKM+1). The box plot for each region has 5 statistics (maximum, upper quartile, median, lower quartile, and minimum, respectively). Abbreviations: AD = artificial diet, AP = apples, PR = pears, PL = plums, PC = peaches and PS = peach shoots.

**Figure 3 insects-13-00893-f003:**
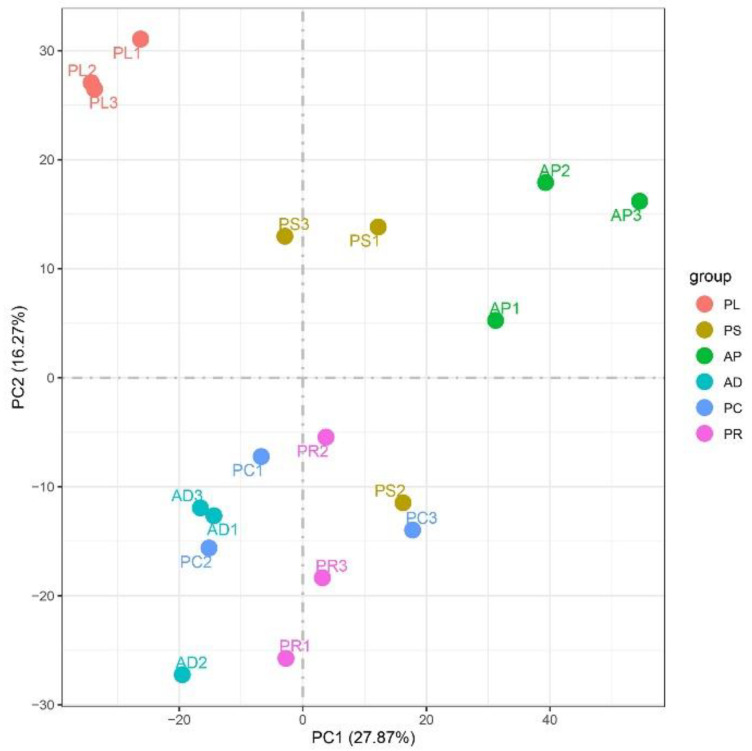
Two-dimensional PCA analysis visualization to measure different host samples. The X-abscissa (PC1) and Y-ordinate (PC2) are the 2 main coordinates with the largest interpretative degree of differences between samples. The same color represents the same grouping. A point is a sample, and similar samples are gathered.

**Figure 4 insects-13-00893-f004:**
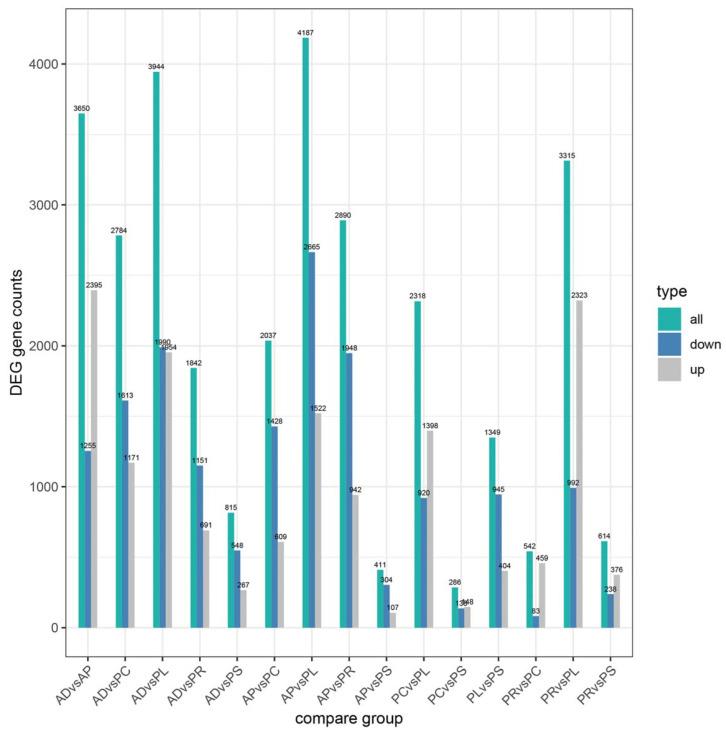
Numbers of upregulated and downregulated DEGs of different hosts by pairwise comparisons.

**Figure 5 insects-13-00893-f005:**
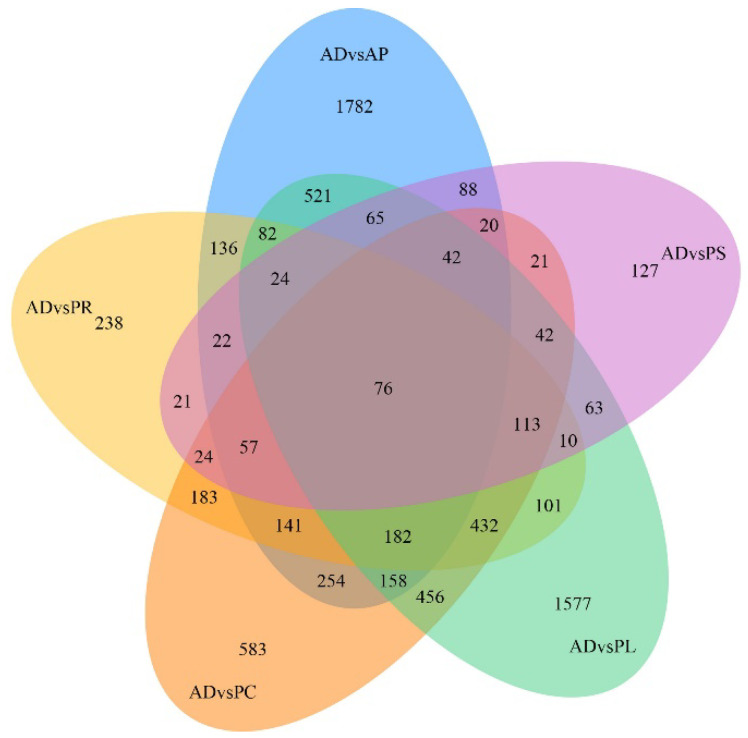
Venn diagram of gene numbers and distribution of 5 different hosts pairwise compared with AD.

**Figure 6 insects-13-00893-f006:**
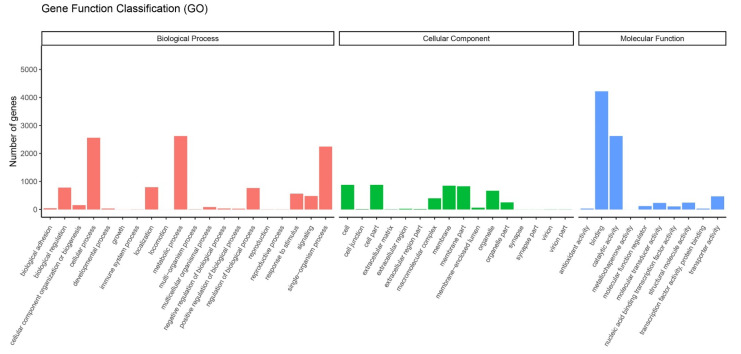
Gene ontology (GO) enrichment analysis of DEGs. GO items were divided into 3 groups: biological processes, cellular components, and molecular functions.

**Figure 7 insects-13-00893-f007:**
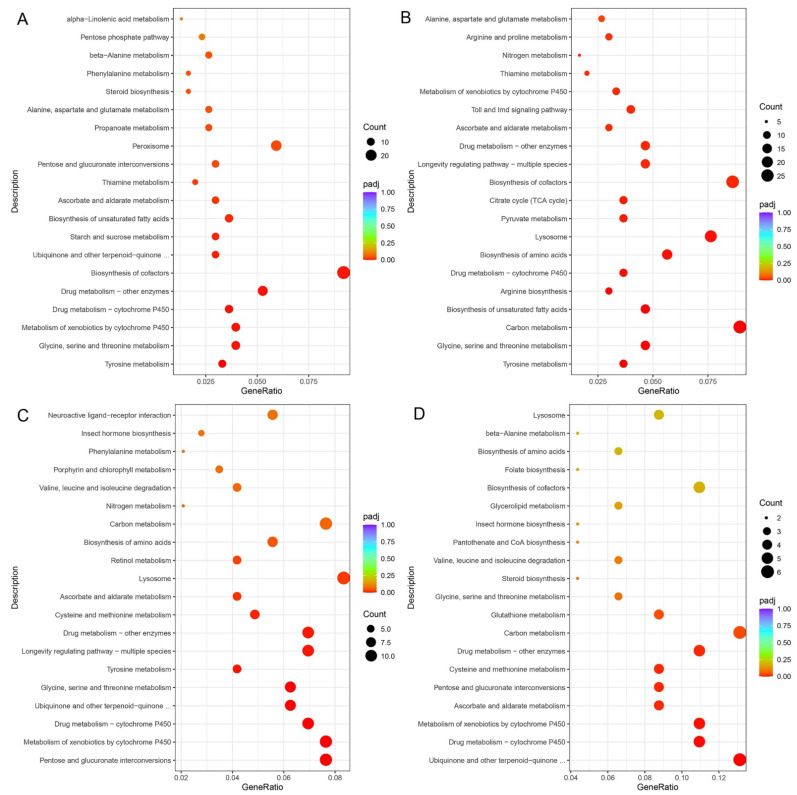
KEGG enrichment scatter plots of DEGs. (**A**). Up-regulated KEGG enrichment pathways of DEGs pairwise of AD vs. PL. (**B**). Up-regulated KEGG enrichment pathways of DEGs pairwise of PR vs. PL. (**C**). Down-regulated KEGG enrichment pathways of DEGs pairwise of PL vs. PS. (**D**). Down-regulated KEGG enrichment pathways of DEGs pairwise of PR vs. PS.

**Figure 8 insects-13-00893-f008:**
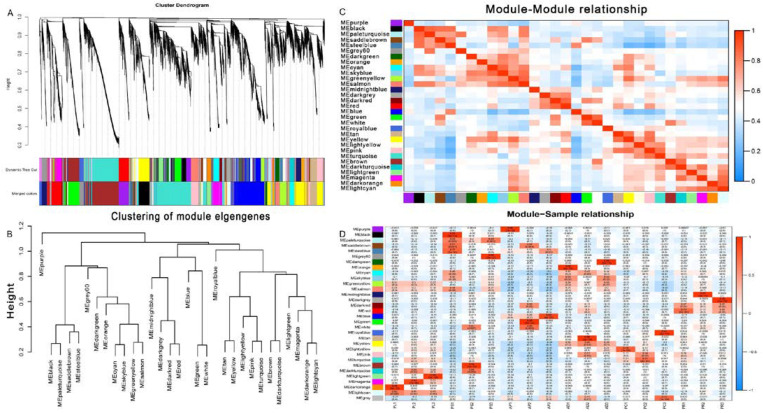
Weighted gene correlation network analysis (WGCNA) and module detection. (**A**) Cluster dendrogram and module assignment for modules from WGCNA. (**B**,**C**) Meta-module identification and module–module relationship. The module network dendrogram was constructed by clustering module eigengene distances. (**D**) The expression patterns of modules are shown as a heatmap. The color bar indicates expression levels of genes from high (red) to low (blue).

**Figure 9 insects-13-00893-f009:**
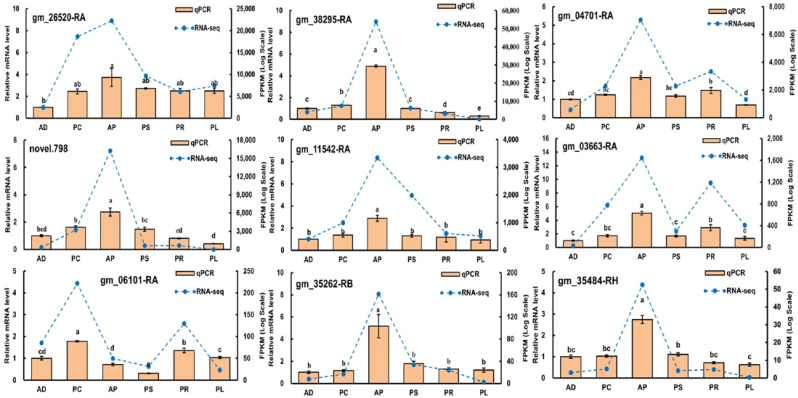
Validation of gene expression by qRT-PCR of selected genes. Different lowercase letters indicate significant differences among treatments (*p* < 0.05). Orange bars indicate the relative expression level (Y-ordinate on left) and the blue lines represent the FPKM values (Y-ordinate on right).

**Table 1 insects-13-00893-t001:** Mean duration (±SE) of developmental stages of *G. molest**a* fed on different hosts.

Host	Larval	Pupa	Adult	Larva–Adult	Egg
AD	15.11 ± 0.57 d	7.57 ± 0.32 b	13.56 ± 0.73 c	36.24 ± 0.11 b	3.60 ± 0.47 b
PC	14.26 ± 0.20 f	7.07 ± 0.44 cd	15.05 ± 0.47 a	36.38 ± 0.72 b	3.13 ± 0.09 d
AP	14.62 ± 0.24 e	6.96 ± 0.25 d	14.13 ± 0.54 b	35.71 ± 0.72 c	3.29 ± 0.32 c
PS	16.42 ± 0.54 a	8.04 ± 0.42 a	13.08 ± 0.22 d	37.55 ± 0.59 a	3.79 ± 0.18 a
PR	15.40 ± 0.38 c	7.17 ± 0.35 c	13.58 ± 0.15 c	36.15 ± 0.55 b	3.81 ± 0.27 a
PL	15.81 ± 0.20 b	7.71 ± 0.15 b	14.05 ± 0.49 b	37.56 ± 0.52 a	3.77 ± 0.18 a

Egg indicates the egg stage of first generation. Means in the same row followed by different lowercase letters differed significantly (*p* < 0.05).

**Table 2 insects-13-00893-t002:** Growth and development parameters of *G. molesta* fed on different hosts.

Host	Larva Survival Rate	Pupation Rate	Pupal Weight	Emergence Rate	Fecundity
AD	45.10 ± 0.14 b	94.50 ± 0.25 b	9.95 ± 0.09 c	88.00 ± 0.06 b	70.20 ± 0.15 d
PC	49.03 ± 0.18 a	96.72 ± 0.93 a	9.24 ± 0.45 d	82.70 ± 0.17 d	65.53 ± 0.38 e
AP	47.53 ± 0.07 a	96.20 ± 0.29 a	9.81 ± 0.32 c	92.55 ± 0.10 a	85.33 ± 0.44 a
PS	42.08 ± 0.11 d	93.53 ± 0.43 d	10.82 ± 0.19 a	85.04 ± 0.38 c	62.36 ± 0.32 f
PR	44.09 ± 0.24 c	93.81 ± 0.87 cd	10.62 ± 0.44 ab	83.00 ± 0.32 d	73.10 ± 0.26 c
PL	44.51 ± 0.11 bc	94.27 ± 0.88 bc	10.48 ± 0.93 b	83.52 ± 0.07 d	75.24 ± 0.14 b

Fecundity indicates the number eggs of per-female. Means in the same row followed by different lowercase letters differed significantly (*p* < 0.05).

**Table 3 insects-13-00893-t003:** Life table parameters of *G. molesta* fed on different hosts.

Host	R_0_	T	r_m_	λ	t_d_
AD	125.05 ± 0.05 b	38.52 ± 0.29 d	0.13 ± 0.02 a	1.22 ± 0.01 ab	5.95 ± 0.03 d
PC	110.10 ± 0.26 c	39.62 ± 0.23 c	0.12 ± 0.01 b	1.15 ± 0.01 d	6.60 ± 0.01 c
AP	135.03 ± 0.12 a	38.83 ± 0.09 d	0.13 ± 0.01 a	1.23 ± 0.01 a	5.90 ± 0.03 d
PS	95.13 ± 0.03 f	39.89 ± 0.20 bc	0.12 ± 0.00 b	1.18 ± 0.01 bcd	6.85 ± 0.01 a
PR	104.00 ± 0.28 e	40.55 ± 0.21 a	0.12 ± 0.01 b	1.19 ± 0.01 bc	6.66 ± 0.01 bc
PL	106.32 ± 0.17 d	40.20 ± 0.40 ab	0.12 ± 0.01 b	1.17 ± 0.01 cd	6.71 ± 0.01 b

Means in the same row followed by different lowercase letters differed significantly (*p* < 0.05).

## Data Availability

All the associated data are available in the manuscript.

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
