# Peer review of "RNA Sequencing Reveals the Potential Adaptation Mechanism to Different Hosts of Grapholita molesta"

_insects, 2022, doi:10.3390/insects13100893_

Round 1
Reviewer 1 Report
This manuscript performed the growth and development of Grapholita molesta feeding on six hosts including: artificial diet, apple, pear, plum, peach and peach shoots. In addition, transcriptomes of G. molesta feeding on different hosts were compared.
General comments
1. The title “RNA-sequencing reveals the potential adaptation mechanism to different hosts of Grapholita molesta” points out the potential adaptation mechanism. Authors should conclude some mechanisms based on the complicated transcriptome data in the manuscript.
2. P. 2, lines 93-94, “…GO annotation showed the biological and molecular functions of the player involved.” What are players?
3. The experimental insects were fed on different hosts for only one generation. Is it long enough to call adaptation?
4. The results of 3.1 appears to replicate some previously published reports including references number 25, 26 and 45 cited in the manuscript. The major concern is whether this study is sufficiently novel. The authors should consider if this part is essential in this manuscript. Or authors need to describe why repeated the similar experiments.
5. The protease activities of the 4th instar larvae feeding on peach shoots were the lowest shown in Fig. 1. However, the expression levels of 9 trypsin genes of the 4th instar larvae feeding on peach shoots were not always the lowest shown in Fig. 9. How do authors explain it?
6. P. 8, line 307, “By comparing the above 18 transcripts…” Do authors mean 18 transcriptomes?
7. P. 13, line 444, authors described that there were 32 modules in fig. 8B. But there were only 31 modules in fig. 8B
8. P. 16, lines 569-570, “Transcriptome analysis confirmed altered gene expression levels in different hosts.” Actually, the gene expression levels are different in G. molesta feeding on different hosts, not in different hosts.
Editing comments
1. P. 1, line 28, peach and peach shoots
2. The resolutions of all 9 figures are not high. It is hard to read.
3. P.7, line 294, delete the "(a, b, c…)"
4. P.15, line 532, replaced “Although 1” with “Although one”.
Author Response
Response to Reviewer 1 Comments
Comments and Suggestions for Authors
This manuscript performed the growth and development of Grapholita molesta feeding on six hosts including: artificial diet, apple, pear, plum, peach and peach shoots. In addition, transcriptomes of G. molesta feeding on different hosts were compared.
General comments
Point 1. The title “RNA-sequencing reveals the potential adaptation mechanism to different hosts of Grapholita molesta” points out the potential adaptation mechanism. Authors should conclude some mechanisms based on the complicated transcriptome data in the manuscript.
Response 1: Has been modified.
In this study, firstly, we found the differences in growth and development parameters and protease activities of different hosts fed by G. molesta. Secondly, transcriptome analysis, including DEGs screening, KEGG enrichment and WGCNA analysis, inferred that trypsin gene plays an important role in the feeding of different hosts. Therefore, qRT-PCR of each host showed that the expression pattern of trypsin gene was consistent with that of RNA-seq. From the differences of growth and development parameters, protease activity characteristics, DEGs screening, enrichment analysis and QRT-PCR were comprehensively revealed that trypsin gene plays an important role in the adaptation to different hosts of G. molesta in many aspects.
Our team focused on the adaptation to different hosts of G. molesta, and this paper mainly included biological parameters and transcriptome data. The results of this study provide a basis for further exploration of the molecular mechanism of host adaptation of G. molesta. More in-depth molecular mechanism has been studied and written papers, the follow-up will be submitted for publication.
Point 2. P. 2, lines 93-94, “…GO annotation showed the biological and molecular functions of the player involved.” What are players?
Response 2: Has been modified. Mistakes in expression and spelling.
Corrected: The differentially expressed genes (DEGs) between feeding hosts were analyzed, and Gene Ontology (GO) annotation showed relevant biological and molecular functions.
Point 3. The experimental insects were fed on different hosts for only one generation. Is it long enough to call adaptation?
Response 3: Lepidopteran larvae show short-term and rapid adaptive responses were evident in host adaptation and host transition on performance, digestive physiology, and intestinal relative gene expression. If the rearing time is too long or reared for many generations, the adaptation of larvae has changed. The following articles are for reference:
- Hafeez M, Li XW, Zhang JM, Zhang ZJ, Huang J, Wang LK, Khan MM, Shah S, Fernández-Grandon GM, Lu YB. Role of digestive protease enzymes and related genes in host plant adaptation of a polyphagous pest, Spodoptera frugiperda. Insect Sci, 2021, 28(3): 611-626. Doi: 10.1111/1744-7917.12906.
- Kumar P, Akhter T, Bhardwaj P, Kumar R, Bhardwaj U, Mazumdar-Leighton S. Consequences of 'no-choice, fixed time' reciprocal host plant switches on nutrition and gut serine protease gene expression in Pieris brassicae L. (Lepidoptera: Pieridae). PLoS ONE, 2021, 16(1): e0245649.
Point 4. The results of 3.1 appears to replicate some previously published reports including references number 25, 26 and 45 cited in the manuscript. The major concern is whether this study is sufficiently novel. The authors should consider if this part is essential in this manuscript. Or authors need to describe why repeated the similar experiments.
Response 4: References number 25.26 and 45 were conducted by Pennsylvania State University in 2006, 2007, and 2013, also more than a decade ago. Pennsylvania State University belongs to North America, and China belongs to Asia. Long-term geographical isolation and differences in living environment inevitably led to some physiological differences among insect populations.
In our experiment, G. molesta were obtained from colony maintain population in the laboratory, and the same batch of eggs were used for the experiment to reduce the error. To study the differences and adaptations of 3 kinds of feeding materials (artificial diet, fruit and peach trees shoots) by G. molesta.
Point 5. The protease activities of the 4th instar larvae feeding on peach shoots were the lowest shown in Fig. 1. However, the expression levels of 9 trypsin genes of the 4th instar larvae feeding on peach shoots were not always the lowest shown in Fig. 9. How do authors explain it?
Response 5: Figure 1 shows the detection of three protease activities on different hosts and artificial diets of G. molesta. It is concluded that the protease activity of feeding on different host and artificial diet is significantly different. Figure 9 comparison with RNA-seq and qRT-PCR analysis of feeding different host and artificial diet of G. molesta. It was concluded that the qRT-PCR of trypsin gene were consistent with RNA-seq in the feeding of different hosts and artificial diets. These results indicate the differential expression of trypsin gene in feeding on different hosts and its role in digestion. The purpose of the two figures is different.
Point 6. P. 8, line 307, “By comparing the above 18 transcripts…” Do authors mean 18 transcriptomes?
Response 6: Has been modified. Mistakes in spelling.
Corrected: By comparing the above 18 transcriptomes with the reference genome, we obtained details of the read map.
Point 7. P. 13, line 444, authors described that there were 32 modules in fig. 8B. But there were only 31 modules in fig. 8B
Response 7: Has been modified. Mistakes in expression. There are 31 modules in Fig. 8 of this paper.
Point 8. P. 16, lines 569-570, “Transcriptome analysis confirmed altered gene expression levels in different hosts.” Actually, the gene expression levels are different in G. molesta feeding on different hosts, not in different hosts.
Response 8: Has been modified.
Corrected: Transcriptome analysis confirmed altered gene expression levels feeding on different hosts.
Editing comments
Point 1. P. 1, line 28, peach and peach shoots.
Response 1: Has been modified.
Point 2. The resolutions of all 9 figures are not high. It is hard to read.
Response 2: Has been modified. All photos in the manuscript have been replaced with high-fidelity resolution photos. The compressed package has been uploaded. Please look at it.
Point 3. P.7, line 294, delete the "(a, b, c…)"
Response 3: Has been modified.
Point 4. P.15, line 532, replaced “Although 1” with “Although one”.
Response 4: Has been modified.

Reviewer 2 Report
The manuscript “RNA-sequencing reveals the potential adaptation mechanism to different hosts of Grapholita molesta”, by Lu et al studied that enrichment and depletion of transcripts in G. molesta on different hosts and artificial diet. The study is comprehensive and can be published after following suggestions are incorporated in the manuscript.
Minor changes
1. Line 14: Delete “which feeds on hosts extensively and does serious harm.”
2. Line 14: Replace “It mainly including” with “Its hosts mainly include”
3. Line 25: Replace” worldwide fruit tree” with “fruit tree worldwide”
4. Line 28: Replace “6 different hosts (artificial diet, apple, pear, plum, peach and peach)” with “5 different hosts (apple, pear, plum, peach and peach) and artificial diet”. Artificial diet is not host of the insect so needs to write separately.
5. Line 43: Replace” worldwide fruit tree” with “fruit tree worldwide”
6. Line 45: Replace” Its host range is very wide” with “It has wide host range”
7. Line 46: Replace “leaves’ shoots” with “leaves, shoots”; and delete “directly feed”
8. Line 46: Either rephrase or delete “double harm to fruit 46 trees, resulting in a large area of fruit damage and”
9. Line 47: Change “loss” to “losses”
10. Line 48: Add word “a” before “characteristic”
11. Line 49: Replace “occurs in” with “has”
12. Line 80: Please replace “of peach” and “of pear” with “on peach” and “on pear”
13. Line 90: As mentioned earlier replace “6 different hosts” with “ 5 hosts and artificial diet”
14. Line 91: Replace “in different hosts of G. molesta” with “of G. molesta feeding on different hosts and artificial diet”
15. Line 104: Remove “the” and add “colony maintained on an” before “artificial”
16. Line 105: Delete “consecutive generations population in the laboratory”
17. Line 106-107: Replace “life cycle” with “generation”
18. Line 108: As mentioned above artificial diet is not host and please rephrase the sentence, “These hosts included artificial diet (AD), apples……………….”
19. Line 121: Replace “amount” with “number of”
20. Line 121: Replace "females” with “female”
21. Line 148: Refer to comment 4, 13, and 18.
22. Line 151: Replace “with” with “using”
23. Line 157: Refer to comment 4, 13, and 18.
24. Line 167-169: Please delete the repeated line.
25. Line 199: Which package of was used?
26. Line 223: Refer to comment 4, 13, and 18.
27. Line 224: Replace “will” with “were”
28. Line 223-224: The author mentioned the most representative combination will (were) selected. Please provide detailed parameters used to determine the combinations that are presented in this study.
29. Line 243: Please replace “obtained” with “extracted”
30. Line 258: Refer to comment 4, 13, and 18.
31. Line 259: Replace “are” with “were”
32. Line 259: Replace “of” with “on”
33. Line 259: Replace “of PS” with “on PS”
34. Line 261-262: Replace “of” with “on”
35. Line 261-264: compare to which treatment? You can say longest and shortest, but when using longer you need to mention compared to which treatment. And relatively shorter is not an apt word. Rephrase these lines.
36. Line 264-265: Significantly different in which treatment? These parameters are not different in all the treatments. Rephrase the sentence that covers all the statistics of the table.
37. Line 265-266: Rephrase the sentence and include the treatment names rather than using other.
38. Line 268: “Life Table parameters of R0, T, rm, λ and td were significantly different” give the impression that that these values were different in all the treatments, which is not true.
39. Line 282 and 290: Add word “instar” after “4th”
40. Line 284, 287, 288, and 290: Replace “of” with “on”
41. Line 307: Change “comparing: with “aligning”
42. Line 325-327: Author mentioned that they bought the fruits from the market but did not mention if they used the same variety. Could the lower R2 be caused by the different varieties if that is the case?
43. Line 349, 353: Refer to comment 4, 13, and 18.
44. Line 354: check the range numbers. They seem off.
45. Line 369, 353: Refer to comment 4, 13, and 18.
46. Line 371: Refer to comment 4, 13, and 18.
47. Line 382 and 396: Please clarify in which treatment the transcriptomes are upregulated, currently the sentence is ambiguous.
48. Figure 8: The color depiction of low is blue not green
49. Line 471: Relative to what?
50. Line 472: “PS is the opposite” is incomplete sentence.
51. Line 477: Replace “degradation” with “decrease”
52. Line 482: AD as mentioned earlier is not host. Rephrase the sentence.
53. Line 485: Does the word “4th” supposed to be here.
54. Line 490: Please support the statement “It may be that the 4rth instar larvae feed a lot and grow faster, so this instar is suitable for a series of experiments.”
55. Line 499: Refer to comment 4, 13, and 18.
56. Line 501: Diet is not a host.
57. Line 502: Replace “feed” with “diet”
58. Line 532: Replace “1” with “one”
59. Line 534: Are you saying “these two hosts” or “the 2nd host”. Please use the word carefully to remove any ambiguity.
60. Line 544: You need to name the enzymes that are enriched rather than the substrates such as sucrose and starch.
61. Line 560: Refer to comment 4, 13, and 18.
62. Line 568: Replace “Asiaticus” with “asiaticus”
Author Response
Response to Reviewer 2 Comments
Comments and Suggestions for Authors
The manuscript “RNA-sequencing reveals the potential adaptation mechanism to different hosts of Grapholita molesta”, by Lu et al studied that enrichment and depletion of transcripts in G. molesta on different hosts and artificial diet. The study is comprehensive and can be published after following suggestions are incorporated in the manuscript.
Minor changes
Point 1. Line 14: Delete “which feeds on hosts extensively and does serious harm.”
Response 1: Has been modified.
Point 2. Line 14: Replace “It mainly including” with “Its hosts mainly include”
Response 2: Has been modified.
Point 3. Line 25: Replace” worldwide fruit tree” with “fruit tree worldwide”
Response 3: Has been modified.
Point 4. Line 28: Replace “6 different hosts (artificial diet, apple, pear, plum, peach and peach)” with “5 different hosts (apple, pear, plum, peach and peach) and artificial diet”. Artificial diet is not host of the insect so needs to write separately.
Response 4: Has been modified.
Point 5. Line 43: Replace” worldwide fruit tree” with “fruit tree worldwide”
Response 5: Has been modified.
Point 6. Line 45: Replace” Its host range is very wide” with “It has wide host range”
Response 6: Has been modified.
Point 7. Line 46: Replace “leaves’ shoots” with “leaves, shoots”; and delete “directly feed”
Response 7: Has been modified.
Point 8. Line 46: Either rephrase or delete “double harm to fruit trees, resulting in a large area of fruit damage and”
Response 8: Has been modified. Has been deleted.
Point 9. Line 47: Change “loss” to “losses”
Response 9: Has been modified.
Point 10. Line 48: Add word “a” before “characteristic”
Response 10: Has been modified.
Point 11. Line 49: Replace “occurs in” with “has”
Response 11: Has been modified.
Point 12. Line 80: Please replace “of peach” and “of pear” with “on peach” and “on pear”
Response 12: Has been modified.
Point 13. Line 90: As mentioned earlier replace “6 different hosts” with “5 hosts and artificial diet”
Response 13: Has been modified.
Point 14. Line 91: Replace “in different hosts of G. molesta” with “of G. molesta feeding on different hosts and artificial diet”
Response 14: Has been modified.
Point 15. Line 104: Remove “the” and add “colony maintained on an” before “artificial”
Response 15: Has been modified.
Point 16. Line 105: Delete “consecutive generations population in the laboratory”
Response 16: Has been modified.
Point 17. Line 106-107: Replace “life cycle” with “generation”
Response 17: Has been modified.
Point 18. Line 108: As mentioned above artificial diet is not host and please rephrase the sentence, “These hosts included artificial diet (AD), apples……………….”
Response 18: Has been modified.
Point 19. Line 121: Replace “amount” with “number of”
Response 19: Has been modified.
Point 20. Line 121: Replace "females” with “female”
Response 20: Has been modified.
Point 21. Line 148: Refer to comment 4, 13, and 18.
Response 21: Has been modified.
Point 22. Line 151: Replace “with” with “using”
Response 22: Has been modified.
Point 23. Line 157: Refer to comment 4, 13, and 18.
Response 23: Has been modified.
Point 24. Line 167-169: Please delete the repeated line.
Response 24: Has been modified.
Point 25. Line 199: Which package of was used?
Response 25: Has been modified.
Point 26. Line 223: Refer to comment 4, 13, and 18.
Response 26: Has been modified.
Point 27. Line 224: Replace “will” with “were”
Response 27: Has been modified.
Point 28. Line 223-224: The author mentioned the most representative combination will (were) selected. Please provide detailed parameters used to determine the combinations that are presented in this study.
Response 28: Has been modified.
Point 29. Line 243: Please replace “obtained” with “extracted”
Response 29: Has been modified.
Point 30. Line 258: Refer to comment 4, 13, and 18.
Response 30: Has been modified.
Point 31. Line 259: Replace “are” with “were”
Response 31: Has been modified.
Point 32. Line 259: Replace “of” with “on”
Response 32: Has been modified.
Point 33. Line 259: Replace “of PS” with “on PS”
Response 33: Has been modified.
Point 34. Line 261-262: Replace “of” with “on”
Response 34: Has been modified.
Point 35. Line 261-264: compare to which treatment? You can say longest and shortest, but when using longer you need to mention compared to which treatment. And relatively shorter is not an apt word. Rephrase these lines.
Response 35: Has been modified.
Point 36. Line 264-265: Significantly different in which treatment? These parameters are not different in all the treatments. Rephrase the sentence that covers all the statistics of the table.
Response 36: Has been modified.
Point 37. Line 265-266: Rephrase the sentence and include the treatment names rather than using other.
Response 37: Has been modified.
Point 38. Line 268: “Life Table parameters of R0, T, rm, λ and td were significantly different” give the impression that that these values were different in all the treatments, which is not true.
Response 38: Has been modified.
Point 39. Line 282 and 290: Add word “instar” after “4th”
Response 39: Has been modified.
Point 40. Line 284, 287, 288, and 290: Replace “of” with “on”
Response 40: Has been modified.
Point 41. Line 307: Change “comparing” with “aligning”
Response 41: Has been modified.
Point 42. Line 325-327: Author mentioned that they bought the fruits from the market but did not mention if they used the same variety. Could the lower R2 be caused by the different varieties if that is the case?
Response 42: Use the same variety for the same fruit. Shown in Section 2.1. These hosts included apples (Malus pumila M, cultivar Fushi, AP), pears (Pyrus bretschneideri R, cultivar Dangshan, PR), plums (Prunus salicina L, cultivar Jujin, PL), and peaches (Prunus persica L, cultivar Qingyan, PC).
Point 43. Line 349, 353: Refer to comment 4, 13, and 18.
Response 43: Has been modified.
Point 44. Line 354: check the range numbers. They seem off.
Response 44: Has been modified.
Point 45. Line 369, 353: Refer to comment 4, 13, and 18.
Response 45: Has been modified.
Point 46. Line 371: Refer to comment 4, 13, and 18.
Response 46: Has been modified.
Point 47. Line 382 and 396: Please clarify in which treatment the transcriptomes are upregulated, currently the sentence is ambiguous.
Response 47: The treatment group are shown in the text. Line 381 to 382 is between AD and PL. Line 395 to 396 is between PR and PL.
Point 48. Figure 8: The color depiction of low is blue not green
Response 48: Has been modified.
Point 49. Line 471: Relative to what?
Response 49: Has been modified.
Point 50. Line 472: “PS is the opposite” is incomplete sentence.
Response 50: Has been modified.
Point 51. Line 477: Replace “degradation” with “decrease”
Response 51: Has been modified.
Point 52. Line 482: AD as mentioned earlier is not host. Rephrase the sentence.
Response 52: Has been modified.
Point 53. Line 485: Does the word “4th” supposed to be here.
Response 53: Has been modified.
Point 54. Line 490: Please support the statement “It may be that the 4rth instar larvae feed a lot and grow faster, so this instar is suitable for a series of experiments.”
Response 54: Has been modified.
Point 55. Line 471: Line 499: Refer to comment 4, 13, and 18.
Response 55: Has been modified.
Point 56. Line 471: Line 501: Diet is not a host.
Response 56: Has been modified.
Point 57. Line 502: Replace “feed” with “diet”
Response 57: Has been modified.
Point 58. Line 532: Replace “1” with “one”
Response 58: Has been modified.
Point 59. Line 534: Are you saying “these two hosts” or “the 2nd host”. Please use the word carefully to remove any ambiguity.
Response 59: Has been modified.
Point 60. Line 544: You need to name the enzymes that are enriched rather than the substrates such as sucrose and starch.
Response 60: Has been modified.
Point 61. Line 560: Refer to comment 4, 13, and 18.
Response 61: Has been modified.
Point 62. Line 568: Replace “Asiaticus” with “asiaticus”
Response 62: Has been modified.

Round 2
Reviewer 1 Report
1. Line 14, Replace “It hosts mainly include” with “Its hosts mainly include”
2. Lines 227-229, There are two sentences mentioned “as representatives of this type”. What is “this type? Please clarify it.
3. Line 268, Replace “Ps” with “PS”
4. Line 268, “The larval-adult stage on PL is the longest.” Based on Table 1, the larva-adult stage on PL is 37.56±0.52, while it is 37.55±0.59 on PS. The statistical data did not support what authors described. And the similar situation happened on the description of the data from Table 3. Please make a reasonable description based on the data.
Author Response
Response to Reviewer Comments
Point 1. Line 14, Replace “It hosts mainly include” with “Its hosts mainly include”
Response 1: Has been modified.
Point 2. Lines 227-229, There are two sentences mentioned “as representatives of this type”. What is “this type? Please clarify it.
Response 2: Has been modified.
Point 3. Line 268, Replace “Ps” with “PS”
Response 3: Has been modified.
Point 4. Line 268, “The larval-adult stage on PL is the longest.” Based on Table 1, the larva-adult stage on PL is 37.56±0.52, while it is 37.55±0.59 on PS. The statistical data did not support what authors described. And the similar situation happened on the description of the data from Table 3. Please make a reasonable description based on the data.
Response 4: Has been modified.
